# Evaluating the Clinical Relevance of Routine Sonication for Periprosthetic Hip or Knee Joint Infection Diagnosis

**DOI:** 10.3390/antibiotics13040366

**Published:** 2024-04-17

**Authors:** Anas Zouitni, Jakob van Oldenrijk, P. Koen Bos, Peter D. Croughs, Erlangga Yusuf, Ewout S. Veltman

**Affiliations:** 1Department of Orthopedic Surgery and Sports Medicine, Erasmus Medical Center, dr. Molenwaterplein 40, 3015 GD Rotterdam, The Netherlands; 2Department of Medical Microbiology and Infectious Diseases, Erasmus Medical Center, dr. Molenwaterplein 40, 3015 GD Rotterdam, The Netherlands

**Keywords:** periprosthetic joint infection (PJI), implant-related infections (IRI), sonication fluid, routine sonication, microbiology, diagnosis, revision arthroplasty, hip arthroplasty, knee arthroplasty, European Bone and Joint Society

## Abstract

Periprosthetic joint infection (PJI) is a serious complication after joint arthroplasty. PJI screening and conventional cultures may be inconclusive. Sonication fluid culturing stands out as a valuable adjunct technique for PJI diagnosis. This study aims to determine the clinical relevance of routine sonication for all (a)septic revisions. All patients who underwent (partial) hip or knee revision arthroplasty between 2012 and 2021 were retrospectively reviewed. We formed three groups based on the European Bone and Joint Society PJI criteria: infection confirmed, likely, and unlikely. We analyzed clinical, laboratory, and radiological screening. Sensitivity and specificity were calculated for synovial fluid (preoperative), tissue, and sonication fluid cultures. We determined the clinical relevance of sonication as the percentage of patients for whom sonication confirmed PJI; 429 patients who underwent (partial) revision of hip or knee arthroplasty were included. Sensitivity and specificity were 69% and 99% for synovial fluid cultures, 76% and 92% for tissue cultures, and 80% and 89% for sonication fluid cultures, respectively. Sonication fluid cultures improved tissue culture sensitivity and specificity to 83% and 99%, respectively. In 11% of PJIs, sonication fluid cultures were decisive for diagnosis. This is applicable to acute and chronic infections. Sonication fluid cultures enhanced the sensitivity and specificity of PJI diagnostics. In 11% of PJI cases, causative pathogens were confirmed by sonication fluid culture results. Sonication fluid culture should be performed in all revision arthroplasties.

## 1. Introduction

The demand for primary and revision knee and hip arthroplasty is increasing, influenced by life expectancy and patient comorbidities [1]. Periprosthetic joint infection (PJI) is a problematic complication and has become one of the most common causes for revision [2,3]. In the literature, the reported infection rates range from 1% to 3% in primary knee and hip arthroplasties [4].

Diagnosing PJI can be challenging. PJI can present itself with clear (acute) signs of infection or with a gradual onset of symptoms. There are multiple definitions with criteria for diagnosing PJI [5,6,7]. In 2021, the European Bone and Joint Infection Society (EBJIS) published an updated definition of diagnostic PJI criteria [6]. The criteria include clinical, laboratory, and radiological findings [6]. 

Identification of PJIs and their causative microorganisms is important to guide treatment planning and improve patient-related outcomes [8]. Early and accurate identification of PJI is necessary to prevent or reduce infection manifestation. Missing or undertreating PJI may cause the persistence of the infection, consequently leading to greater patient morbidity, mortality, and hospital costs [9,10]. PJI management strategies usually consist of surgery followed by antimicrobial therapy. Surgical interventions include debridement and implant retention (DAIR), a one-stage revision, a two-stage revision, (permanent) removal of the prosthesis (Girdlestone), or amputation. Antimicrobial therapy is determined based on the pathogen susceptibility and patient status [7,10]. 

Infection can be confirmed by positive (preoperative) synovial fluid and tissue cultures [5,6,7]. Synovial fluid cultures have a sensitivity of 72–84% and a specificity of 93–98% [11]. The clinical features of a sinus tract (with communication to the joint), or visualization of the prosthesis, can confirm infection. Histological analysis and/or synovial fluid analysis (leukocyte count, polymorphonuclear neutrophil percentage (PMN), and positive alpha defensin) can also confirm or refute suspicion of infection [6]. Tissue cultures have a sensitivity of 66–94% and a specificity of 67–99% [12,13,14,15,16]. Due to the low sensitivity of synovial fluid cultures and tissue cultures, some of the causative microorganisms remain unidentified. As a result, many studies describe a proportion of culture-negative PJIs ranging from 5% to 42% [17].

Chronic infections with low virulent pathogens that produce biofilms are difficult to detect by synovial and tissue cultures [18]. Sonication fluid cultures can be used as an adjunct diagnostic tool alongside synovial fluid and tissue cultures. Sonication is reported to dismantle biofilms on prostheses using ultrasound, releasing bacteria in the surrounding fluid. This offers the possibility of identifying a broader range of pathogens [19]. Positive sonication fluid cultures yield results faster than tissue cultures in chronic infections [20]. The literature on sonication reports a sensitivity ranging from 47 to 97% and specificity from 90 to 100% [12,13,14,15,16]. The combination of tissue and sonication fluid culture has been reported to reach a higher sensitivity of definite microbiological diagnosis [12]. Even a single positive sonication fluid culture may indicate a potential infection or confirm infection [6,21]. Since February 2012, we have routinely used sonication for all (a)septic revisions.

Routine sonication is not widely used due to concerns about its clinical relevancy, labor intensity, and costs. Adequate information on how routine sonication influences clinical management and treatment is lacking. Most of the studies on this topic are limited to small study cohorts. This may not provide a comprehensive understanding, indicating the need for further research.

This study aims to evaluate the clinical relevance of routinely using sonication fluid cultures for PJI diagnosis in all (a)septic hip and knee revision arthroplasties. We want to identify how often sonication fluid cultures are decisive for the microbiological diagnosis of PJI. 

## 2. Results

### 2.1. Demographics

We included 429 patients who underwent (partial) revision hip or knee arthroplasty at our clinic between 2012 and 2021. Table 1 lists the (preoperative) demographics of all the included patients. Postoperative, 110 cases were determined to have PJI, and 319 cases were determined to be aseptic.

### 2.2. Sensitivity and Specificity

Sensitivity and specificity were 69% and 99% for synovial fluid cultures, 76% and 92% for tissue cultures, and 80% and 89% for sonication cultures, respectively. The results of our study on sensitivity and specificity can be found below in Table 2. Sonication improved tissue culture sensitivity and specificity to 83% and 99%. 

### 2.3. Subgroup Analysis

We performed a subgroup analysis on 234 patients who had all three culture types available for direct comparison (Table 3). We found no significant difference in sensitivity between sonication fluid cultures versus tissue cultures (81% [70–89%] versus 76% [65–85%], with *p* = 0.32 (>0.05)). Specificity did not significantly differ between the sonication fluid cultures versus the tissue cultures (90% [85–94%] versus 95% [91–98%], with *p* = 0.09). We found a significant difference between the sensitivity and specificity of the sonication fluid cultures and the preoperative synovial fluid cultures. Sensitivity for sonication was 81%; [70–89%], compared to preoperative synovial fluid; 68% [57–79%] with *p* = 0.03 (<0.05). Specificity for sonication was 90% [85–94%], compared to preoperative synovial fluid; 99% [97–100%] with *p* < 0.001.

### 2.4. The Clinical Relevance of Routine Sonication

We identified 12 (11%) out of 110 cases of PJI where sonication was the determining factor for confirming the causative pathogen(s). Nine of these 12 patients were in the infection-likely group, and three were in the infection-confirmed group. For the nine patients in the infection-likely group, sonication changed the definite diagnosis to infection confirmed. For the three patients in the infection-confirmed group, even though PJI was already confirmed, sonication was decisive for the microbiological PJI diagnosis. All 12 cases where sonication was decisive had a positive sonication fluid culture, with one positive tissue or preoperative synovial fluid culture, with identical microorganisms. Among these 12 cases, six cases were chronic infections, and the other six cases were acute (hematogenous) infections (Table 4). We found *Staphylococcus aureus* and *Staphylococcus epidermidis* (CNS) in three cases each, *Staphylococcus haemolyticus* (CNS) in two cases, and *Streptococcus dysgalactiae* (beta-hemolytic streptococcus group C), *Listeria monocytogenes*, *Klebsiella oxytoca*, *Escherichia coli*, and *Haemophilus parainfluenzae* in one case each. For other confirmed PJIs, we found multiple positive tissue cultures or matching positive preoperative synovial fluid and tissue cultures.

### 2.5. Detected Microorganisms

The microorganisms most frequently detected by sonication were different for confirmed PJI versus aseptic (Table 5). *Staphylococcus aureus* and *Staphylococcus epidermidis* (CNS) were most often detected in cases of confirmed PJI. In total, sonication fluid detected a broader variety of microorganisms. Preoperative synovial fluid, tissue, and sonication fluid cultures detected 28, 35, and 40 different microorganisms, respectively. In 12 (11%) out of 110 PJI cases, sonication could confirm contaminants of false-positive preoperative synovial fluid and tissue cultures. Conversely, 12 (11%) cases in the same group had negative sonication fluid cultures whilst infection was confirmed. In 29 (9%) out of 319 aseptic cases, sonication could confirm contamination of tissue cultures. In the same group, 30 (10%) cases were determined aseptic, while sonication was false positive (contamination).

## 3. Discussion

We aimed to evaluate the clinical relevance of routine sonication fluid cultures for diagnosing PJI. Sonication fluid culture results confirmed a microbiological diagnosis of PJI for 12 (11%) out of 110 patients. They would have been misdiagnosed or undertreated based on preoperative synovial fluid and tissue cultures alone. Routine sonication identified the causative pathogen in presumed septic cases (infection likely) and preoperative confirmed PJI cases. In 12 (11%) out of 110 PJI cases, sonication could confirm contaminants of preoperative synovial fluid and tissue cultures. In 29 (9%) of the aseptic cases, negative sonication could confirm contaminants of (preoperative) synovial fluid cultures and tissue cultures. This indicates an increase in the negative predictive value. Sonication fluid culturing has an added clinical value for all patients (septic and aseptic). In the PJI group, 12 (11%) cases had negative sonication fluid cultures, whilst infection was confirmed. This indicates a decrease in the positive predictive value. However, since PJI diagnosis does not solely depend on sonication, and PJI was confirmed by other factors, this is not clinically relevant.

We evaluated the sensitivity and specificity of preoperative synovial fluid, tissue, and sonication fluid cultures. We found a sensitivity of 80% and a specificity of 89% for sonication. Although not statistically significant, sonication fluid cultures demonstrated a higher sensitivity than tissue cultures. Sonication fluid cultures increased the sensitivity and specificity of tissue cultures to 83% and 99%, respectively. Preoperative synovial fluid cultures have a significantly lower sensitivity compared to sonication fluid cultures. 

Low virulent microorganisms may be difficult to detect by synovial fluid and tissue cultures, resulting in false-negative results or false-positive results in cases of contamination [18]. We found that pathogens confirmed by sonication were mostly low virulent (Table 4 and Table 5). In the group where the sonication results were decisive, both virulent and low-virulent pathogens were found. 

The limitations of this study are reflective of its retrospective design. In some cases, we found the information to be incomplete or unclearly written in the medical records. We also performed a subgroup analysis on patients who had all three culture types available. This analysis was prone to selection bias because only 234 of 430 patients were available for direct comparison. The timeframe of our inclusion period could explain the inconsistency of diagnostics, as PJI diagnosis guidelines change over time. Eventually, this inconsistency did not matter since the results for the subgroup and the entire cohort were comparable. We used the current updated EBJIS criteria as a guideline to retrospectively classify our study population [6]. Since it is a consensus definition, it does not provide complete certainty about the presence of PJI.

We have a large study population, which helps to validate earlier findings for future research. Multiple studies describe the value of sonication in diagnosing PJI. We found similar results in our study. A study showed that, in 9% of the patients with PJI, sonication was the determining factor for microbiological diagnosis [14]. In our study, we found that in 11% with PJI, sonication was the determining factor. This highlights the clinical importance of sonication in PJI diagnosis and treatment. 

Anti-biofilm microbiological techniques like sonication can yield faster bacterial growth [20]. Sonication is harmless for the patient and is probably cost-effective since untreated infection may result in higher costs and patient morbidity [22]. The costs for our institute may differ from other institutes. Sonication may also increase labor intensity, which demands additional costs for personnel.

Studies on sonication mainly compare the sensitivity and specificity of sonication to tissue cultures [12,16,23]. Opponents of the use of sonication fluid cultures criticize it for being unreliable because of its low sensitivity in comparison to tissue cultures [16]. These studies fail to evaluate (routine) sonication fluid cultures as an adjunct to tissue cultures. Instead, they evaluate the potential of sonication as a replacement. We assume that the clinical relevance of sonication fluid cultures merits its use, for as much as 11% of PJIs may remain undiagnosed or undertreated without sonication culture results.

The number of positive cultures is important for diagnosis. The literature shows that single positive tissue cultures in presumed aseptic cases may not be clinically relevant. These cases had an equal infection-free implant survival to presumed aseptic cases with negative cultures [24]. This shows that it is questionable to treat aseptic patients with a single positive culture where contamination is more likely. Another study showed that, in aseptic cases, two cultures with identical microorganisms reduced infection-free implant survival [25]. If these cases have low-virulent microorganisms that are difficult to detect with tissue cultures, sonication may provide enough evidence to confirm PJI or refute suspicion. According to the EBJIS PJI criteria, a single positive culture (preoperative synovial fluid or tissue culture) with a virulent organism makes an infection more likely. The criteria also state that any positive sonication fluid culture (above 50 CFU/mL) is a potential infection [6]. A recent study suggests that even a single positive sonication culture may have important implications for treatment [21]. This validates that sonication may also be clinically relevant for unsuspected PJI in aseptic patients. In our study, in the 12 cases where sonication was decisive, we found four cases with *Staphylococcus aureus* as a causative pathogen (Table 4). *Staphylococcus aureus* is a virulent pathogen and is likely to be the cause of some PJIs. In these cases, tissue cultures alone were maybe sufficient to determine infection and that sonication was unnecessary. Further research on this issue is needed.

Future research could also evaluate the use of sonication on other implant-related infections. This could be for materials that are not in touch with synovial fluid, such as after osteosynthesis or spinal fixations. For these materials, microbiological infection confirmation is dependent on tissue cultures and/or sonication fluid cultures alone. This could demonstrate an even more extensive clinical relevance for routine sonication fluid cultures. 

## 4. Materials and Methods

We consulted the STROBE statement while designing the study and writing the manuscript.

### 4.1. Study Population

Sonication fluid culture was performed in all revisions per local protocol, both for septic and aseptic revisions. This retrospective cohort study includes all patients who underwent (partial) revision hip or knee arthroplasty at our clinic between February 2012 and May 2021. Although sonication has been advised as an additional diagnostic tool for (presumed) septic revisions, we have used sonication for all revisions since February 2012. We collected all data from digital medical records. We included the demographic characteristics of age and biological sex (male or female). Patients with unavailable sonication fluid cultures were excluded. 

We classified and divided patients into three groups based on the EBJIS PJI criteria (preoperative): infection confirmed, infection likely, and infection unlikely [6]. We classified patients as ‘infection unlikely’ if they had a clear alternative reason for prosthesis dysfunction and did not meet other EBJIS criteria. All patients received preoperative prophylactic antibiotics 15–60 min before incision. In some cases, patients with high suspicion of PJI or confirmed PJI received empiric intravenous antibiotic treatment. Treatment was given until definite culture results and bacteria susceptibility to antibiotics were available. Based on definite culture results, antibiotic treatment could be continued or discontinued. In presumed aseptic revisions, patients received prophylactic antibiotic treatment for 24 h. We differentiated between acute PJI (less than three months) and chronic PJI (more than three months) based on the duration of symptoms. Acute PJI could also be due to a hematogenous source of infection.

### 4.2. Microbiology

We analyzed the variety of microorganisms found by preoperative synovial fluid, tissue, and sonication fluid cultures. According to the EBJIS PJI criteria, causative pathogens are confirmed by two or more separate cultures yielding the same microorganism. Any positive sonication fluid culture is considered a potential infection. If the culture yields more than 50 colony-forming units/mL (CFU/mL), it can confirm infection [6].

Contamination of microorganisms could occur during the collection and processing of the cultures. Microorganisms identified by a single culture and that are unlikely to cause PJI were defined as contaminants. A single culture with a virulent microorganism (such as *Staphylococcus aureus*) could be either a pathogen or contaminant. In these cases, further investigation was needed to refute suspicion of infection [6]. All culture results were consulted by our microbiology department to differentiate pathogens from contaminants and to establish treatment plans.

### 4.3. Preoperative Cultures

For some cases, preoperative synovial fluid culture and/or leukocyte count were attempted. For knees, aspiration of synovial fluid was performed after skin disinfection and sterile field draping, using a no-touch technique in the outpatient clinic. For hips, aspiration was performed under fluoroscopic guidance, after skin disinfection and field draping, by an orthopedic resident or surgeon. In some cases, when dry tap aspiration with high clinical suspicion of PJI was not sufficient, tissue biopsies were taken in the OR. In cases of dry tap with low suspicion of PJI, the absence of synovial fluid was considered unsuspicious. However, in case of doubt due to clinical suspicion of PJI, tissue biopsies were still taken in the OR. Aseptic patients did not always have aspiration or tissue biopsies taken. In cases of suspected acute infections, aspiration was not regularly performed because a DAIR procedure followed immediately.

### 4.4. Perioperative Cultures

During surgery, the aim was to collect 4 to 6 tissue cultures for microbiology. Tissue samples were taken from different periprosthetic tissue locations. All tissue cultures were taken with separate sterile rongeurs and placed in separate sterile containers. The containers were immediately delivered to the microbiology laboratory for further analysis. Tissue samples were divided into 0.1 mL aliquots and placed onto aerobic blood agar, chocolate agar, McConkey agar, and Brucella blood agar. The aliquots were put into thioglycolate broth. The tissue samples were homogenized in a brain–heart infusion broth. The aerobic cultures were incubated at 37 °C for 48 h.

### 4.5. Sonication-Fluid Cultures 

The prosthetic components that were explanted during surgery were sent to the microbiology laboratory in a sterile polypropylene container. All materials were processed within 6 h after implant removal and were stored at room temperature in the polypropylene container. A sodium chloride solution covered the implant for at least 90%. To process the materials, the container was shaken manually for 30 s. The container was then placed in a sonication bath where sonication takes place for 1 min at 40 kHz. Then, with a pipette, 100 μL of sonication fluid were transferred to three agar plates (blood agar, chocolate agar, and Brucella agar). Ten mL of sonication fluid were inoculated into the aerobe and anaerobe blood-culture bottle with a sterile syringe. The blood agar and chocolate agar were incubated at 35–37 °C and 5% CO_2_ for 14 days. During this period, bacterial growth was assessed daily. The Brucella agar was incubated anaerobically at 35–37 °C for 14 days and assessed for growth at days 2, 5, 7, and 14. In case of growth on the plates, the number of colony-forming units were counted. The blood-culture bottles were incubated for 14 days, and growth was documented as growth after accumulation. The culture was considered negative when no growth occurred both on the plates and in the blood-culture bottles after 14 days of incubation. 

### 4.6. Statistical Analysis

We summarized general patient characteristics using descriptive statistics. Percentages were used for presenting categorical variables. The sensitivity, specificity, negative predictive value, and positive predictive value of the diagnostic tools were calculated using 2 × 2 contingency tables. We used a clinical classification with the EBJIS PJI criteria as a reference standard to determine these values [6]. We considered a culture as positive if the detected microorganism was confirmed. In addition to the culture results, the clinical classification included other factors that confirm PJI, for example, the presence of a sinus tract [6]. We classified the culture techniques as a false negative if they failed to detect the causative pathogen in case of confirmed PJI. The culture was classified as a false positive if it detected a contaminant. We performed McNemar’s test to compare paired data. A *p*-value smaller than 0.05 for a two-sided test was considered statistically significant. We performed the statistical analysis using IBM SPSS Statistics (version 28.0.1.0).

## 5. Conclusions

Our results show that sonication has an added value in PJI diagnostics. Sonication was confirmative for both acute and chronic infections. For aseptic revisions, routine sonication fluid cultures may be helpful to confirm contamination of tissue or preoperative synovial fluid cultures. Due to its clinical relevance in detecting PJI, we believe routine sonication fluid cultures should be performed in all revisions. 

## Figures and Tables

**Table 1 antibiotics-13-00366-t001:** Patient demographics.

Characteristic	Infection Confirmed (n = 58)	Infection Likely (n = 57)	Infection Unlikely (n = 314)
Joint, hip (%)	33 (57%)	35 (61%)	178 (57%)
Joint, knee (%)	25 (43%)	22 (39%)	136 (43%)
Age (years)			
Median	69	70	68
Range	36–87	32–89	27–93
Sex			
Male	29 (50%)	20 (35%)	101 (32%)
Female	30 (50%)	37 (65%)	213 (68%)
Clinical features			
Radiographic loosening, number (%)	30 (52%)	26 (46%)	129 (41%)
Temperature ≥ 38 °C (%)	9 (15%)	10 (17%)	2 (1%)
Purulence around prosthesis (%)	38 (65%)	24 (42%)	0
Sinus tract	36 (62%)	0	0
Visible prosthesis	2 (3%)	0	0
Symptom duration > 3 months, number (%)	37 (64%)	38 (67%)	293 (93%)
Blood workup			
Serum C reactive protein (CRP) ≥ 10 mg/L, number/total number (%)	53 (91%)	43/56 (77%)	96/291 (33%)
Erythrocyte sedimentation rate (ESR) ≥ 30 mm/h, number/total number (%)	47/55 (85%)	37/52 (71%)	65/290 (22%)
Bacteremia, number (%)	5/9 (56%)	7/15 (47%)	0
Synovial fluid cytological analysis			
Leukocyte count (cells/μL) number/total number (%)			
≥1500–<3000 (×10^6^/L)	1/27 (4%)	0	11/86 (13%)
≥3000 (×10^6^/L)	26/27 (96%)	3/13 (23%)	16/86 (19%)
Neutrophil % in synovial fluid,			
>65–<80%	7/28 (25%)	3/11 (27%)	3/59 (5%)
>80%	20/28 (75%)	0	16/59 (27%)
Other			
Nuclear imaging performed, number (%)	13 (22%)	14 (25%)	96 (31%)

**Table 2 antibiotics-13-00366-t002:** Sensitivity, specificity, positive predictive value, and negative predictive value.

Culture Type	Sensitivity	Specificity	Positive Predictive Value	Negative Predictive Value
Preoperative synovial fluid cultures	69%	99%	98%	89%
Tissue cultures	76%	92%	77%	92%
Sonication fluid cultures	80%	89%	72%	93%

**Table 3 antibiotics-13-00366-t003:** Subgroup analysis of patients with synovial fluid (preoperative), tissue, and sonication fluid cultures.

Culture Type	Sensitivity	Specificity
Preoperative synovial fluid cultures	68%	99%
Tissue cultures	76%	95%
Sonication fluid cultures	81%	90%

**Table 4 antibiotics-13-00366-t004:** Cases with positive sonication and only one positive preoperative synovial fluid culture or tissue culture.

Case	Time Since Primary Arthroplasty to Revision (Years)	Symptom Duration > 3 Months	Acute Hematogenous Infection	Causative Pathogens	Previous Treatment for PJI (Antibiotics)
1	5	No	Yes	*Streptococcus dysgalactiae* (hemolytic streptococcus group C)	Yes, (Augmentin I.V. and Amoxicillin)
2	1	Yes	No	*Staphylococcus aureus* and *Staphylococcus epidermidis* (CNS) *	Yes, (Ciprofloxacin and Doxycycline)
3	3	No	Yes	*Staphylococcus haemolyticus* (CNS) *	Yes, (Meropenem and Vancomyocin)
4	14	No	Yes	*Escherichia coli*	No
5	6	Yes	No	*Haemophilus parainfluenzae*	No
6	2	Yes	No	*Staphylococcus haemolyticus* (CNS) *	Yes (Augmentin)
7	6	Yes	No	*Staphylococcus aureus*	Yes (Cefuroxim, Fluxoclacillin, Rifampicin, Levofloxacillin)
8	12	Yes	No	*Staphylococcus epidermidis* (CNS) *	No
9	8	No	Yes	*Staphylococcus aureus*	Yes
10	2	Yes		*Listeria monocytogenes*	No
11	4	No	Yes	*Klebsiella oxytoca*	No
12	14	No	Yes	*Staphylococcus epidermidis* (CNS) *	No

* CNS = coagulase-negative staphylococcus.

**Table 5 antibiotics-13-00366-t005:** Frequency of top-five bacterial species detected by sonication-fluid cultures.

PJI		Aseptic	
Microorganism	Frequency (n)	Microorganism	Frequency (n)
*Staphylococcus aureus*	27	*Cutibacterium acnes*	8
*Staphylococcus epidermidis* (CNS) *	22	*Staphylococcus epidermidis* (CNS) *	7
*Enterococcus faecalis/faecium*	8	*Staphylococcus capitis* (CNS) *	3
*Cornyebacterium striatum*	6	*Micrococcus luteus*	3
*Cutibacterium acnes*	5	*Staphylococcus hominis* (CNS) *, *Staphylococcus species* (CNS) *	2

* CNS = coagulase-negative staphylococcus.

## Data Availability

The original contributions presented in the study are included in the article; further inquiries can be directed to the corresponding author.

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
