# Peer review of "Evaluating the Clinical Relevance of Routine Sonication for Periprosthetic Hip or Knee Joint Infection Diagnosis"

_antibiotics, 2024, doi:10.3390/antibiotics13040366_

Round 1

Reviewer 1 Report

Comments and Suggestions for Authors

Very Respected Authors,

After carefully reading your paper I determined that the abstract is well-written. The introduction provides informative content with a clear objective. The methodology is thoroughly explained. The results are clearly presented, and the discussion is well-written. The conclusion is in agreement with the objective of the paper and with the results.

Author Response

Thank you for your time and your positive review of our manuscript.

Reviewer 2 Report

Comments and Suggestions for Authors

Thank you for an enjoyable article.

Abstract:

Good summary of article - pertinent points related

Introduction:

Background quotes demographics of TJR infection well. Describes the use of sonification as an adjunct to diagnosis - previous studies describe it as a solely used tool. 

The concept of cost and labour intensiveness is touched upon - this is important as fiscal responsibility is also a consideration for our patients and the healthcare system.

Study population:

Retrospective study - numbers of patients and exclusion criteria explained.

Defined acute versus chronic infection in line with accepted definitions

The definition of infection was clear (>2 separate cultures).

Acute infections were treated with DAIR in line with current accepted practice.

Perioperative cultures:

The method described is consistent

Statistical analysis:

Validated definitions used, and the McNemar test is appropriate for the data.

A large sample size was used.

The subgroup analysis allowed for sonification to provide additional information re: infection when paired with cultures.

The microorganisms detected are in line with common pathogens for joint infection world wide.

Discussion and Conclusions:

The authors acknowledge the limitations of a retrospective study and incomplete data.

They discuss the lack of harm to patients in using sonification, and its use as an adjunct to cultures, expanding on earlier sections.

Future research paths are also discussed.

The conclusions are soundly argued.

Author Response

(The authors gave the same response as above.)

Reviewer 3 Report

Comments and Suggestions for Authors

- my recommendation would be for the authors to expand on the introduction section. I would include the importance of efficient and early PJI diagnosis and its management strategies

- based on the research done so far, authors should also include how sonication improved diagnostic accuray, patient outcomes or other healthcare institutional related outcomes

- can the authors provide a reason of inclusion of all patients undergoing revision arthroplasty without segregating them into classes such as: aseptic vs septic? This could be a major limitation if the reason is not clearly specified.

- this is a high value study. Authors mention that they used Microsoft Excel for statistical analysis. To my knowledge, this does not allow for advanced statistical tests and variance calculations. Can the authors elaborate on this?

- what are the biases that authors consider that apply to their study design and manuscript?

- my suggestion is that authors clearly define some research questions at the end of their discussion section

- the discussion section should take a more neutral approach towards sonication, supporting both advantages and disadvantages

- the title is adequate

- references are up to date (can be improved if suggestions above are met)

- study design is adequate

- results elaboration and presentation are adequate

Author Response

Thank you for your time and your remarks. For clarity issues, we will answer each one-by-one below, with our answers in +green.

- my recommendation would be for the authors to expand on the introduction section. I would include the importance of efficient and early PJI diagnosis and its management strategies.
+ Thank you for this remark. In lines 45-52, we have now included information on the importance of PJI diagnosis and its management strategies.

- based on the research done so far, authors should also include how sonication improved diagnostic accuracy, patient outcomes or other healthcare institutional related outcomes
+ Thank you for his remark. In lines 63-72 we have elaborated on the current state of the literature on sonication. We think that this part of the manuscript already outlines its improved accuracy in detecting specific infections. If the editor feels that we should further expand, we could do this. However, the authors think that this would not add value to the message of the manuscript.
We have included how much sonication improved sensitivity and specificity in our results section (lines 212-215).

- can the authors provide a reason of inclusion of all patients undergoing revision arthroplasty without segregating them into classes such as: aseptic vs septic? This could be a major limitation if the reason is not clearly specified.
+ Thank you for the comment. We have tried to clarify this in lines 94-96, by removing the word ‘(a)septic’. Based on discussions in our multidisciplinary meeting, discussing the options for reducing the number of ‘culture negative’ infections, we decided in 2012 to perform sonication on all our revision cases, instead of only a selection.

- this is a high value study. Authors mention that they used Microsoft Excel for statistical analysis. To my knowledge, this does not allow for advanced statistical tests and variance calculations. Can the authors elaborate on this?
+ Thank you for this remark, you are absolutely right. We started with using Excel for data collection. Later, we transferred this information in an IBM SPSS Statistics database which allows advanced statistical tests and variance calculations. We have adjusted this in the manuscript (line 182-183).

- what are the biases that authors consider that apply to their study design and manuscript?
+ Thank you for the question. We have elaborated on the limitations of this study in lines 323-332. The biases introduced in this study are typically for its retrospective design. Firstly, we mentioned that our subgroup analysis is prone to selection bias. Secondly, we mentioned that information was incomplete or unclearly written down. This could be the result of information bias. However, incomplete data it is more likely to be the result of the timeframe of inclusion. From 2012 to 2021, guidelines and protocols have changed and may have influenced the type of screening method. Furthermore, we got our information through medical records which is the most reliable source for our data.

- my suggestion is that authors clearly define some research questions at the end of their discussion section
+ Thank you for your suggestion. We feel that sonication has proved its value for detection of PJI. A cost-effectiveness study could and maybe should however still be performed, but prospective studies evaluating details such as the value of sonication require such a vast number of patients that these studies are hardly executable. In lines 341-343, we have added that cost-effectiveness still has to be evaluated for sonication and may differ per institution. On the other hand, we feel it is more important to first study if sonication is clinically relevant in other infections than arthroplasty, we have advised studying this in lines 369-373.

- the discussion section should take a more neutral approach towards sonication, supporting both advantages and disadvantages
+Thank you for this remark. We think it is important to keep the advantages in the discussion section. We have now added more disadvantages to make it more neutral: negative sonication fluid cultures in confirmed PJI in lines 308-311, and of costs and labor intensity of implementing sonication in lines 341-343.